# 14-3-3 Proteins and the Plasma Membrane H^+^-ATPase Are Involved in Maize (*Zea mays*) Magnetic Induction

**DOI:** 10.3390/plants12152887

**Published:** 2023-08-07

**Authors:** Anna Fiorillo, Ambra S. Parmagnani, Sabina Visconti, Giuseppe Mannino, Lorenzo Camoni, Massimo E. Maffei

**Affiliations:** 1Department of Biology, Tor Vergata University of Rome, Via della Ricerca Scientifica, 00133 Rome, Italy; anna.fiorillo@uniroma2.it (A.F.); sabina.visconti@uniroma2.it (S.V.); 2Department of Life Sciences and Systems Biology, University of Turin, Via Quarello 15/a, 10135 Turin, Italy; ambraselene.parmagnani@unito.it (A.S.P.); giuseppe.mannino@unito.it (G.M.)

**Keywords:** geomagnetic field, near-null magnetic field, coleoptiles and roots, transcriptomics, phytohormones, IAA, *trans*-zeatin, melatonin, iron-sulfur complex assembly, plant magnetoreception

## Abstract

The geomagnetic field (GMF) is a natural component of the biosphere, and, during evolution, all organisms experienced its presence while some evolved the ability to perceive magnetic fields (MF). We studied the response of 14-3-3 proteins and the plasma membrane (PM) proton pump H^+^-ATPase to reduced GMF values by lowering the GMF intensity to a near-null magnetic field (NNMF). Seedling morphology, H^+^-ATPase activity and content, 14-3-3 protein content, binding to PM and phosphorylation, gene expression, and ROS quantification were assessed in maize (*Zea mays*) dark-grown seedlings. Phytohormone and melatonin quantification were also assessed by LG-MS/MS. Our results suggest that the GMF regulates the PM H^+^-ATPase, and that NNMF conditions alter the proton pump activity by reducing the binding of 14-3-3 proteins. This effect was associated with both a reduction in H_2_O_2_ and downregulation of genes coding for enzymes involved in ROS production and scavenging, as well as calcium homeostasis. These early events were followed by the downregulation of IAA synthesis and gene expression and the increase in both cytokinin and ABA, which were associated with a reduction in root growth. The expression of the homolog of the MagR gene, *ZmISCA2*, paralleled that of *CRY1*, suggesting a possible role of ISCA in maize magnetic induction. Interestingly, melatonin, a widespread molecule present in many kingdoms, was increased by the GMF reduction, suggesting a still unknown role of this molecule in magnetoreception.

## 1. Introduction

One of the most hidden abiotic plant stress factors in the Earth’s biosphere is the variation in the intensity and inclination of the geomagnetic field (GMF) [1]. The GMF, which is an environmental component of Earth, has always been present since the beginning of plant evolution [2]. One of the key roles of the GMF is to protect Earth’s living organisms against solar wind and cosmic radiation, which may alter the planet’s climate and atmosphere, as well as affect several cell functions [3]. Moreover, the GMF could have influenced the origin [3] and evolution [4] of life. To study the effect of the GMF on plant biology, we built an exposure system made of a triaxial Helmholtz coil system able to reduce the GMF (with an intensity of about 43 µT) to near null magnetic field (NNMF) values (of about 30 nT; i.e., the typical values of the cosmic magnetic field) [5]. In a series of experiments, we demonstrated that plants differentially perceive magnetic field (MF) variations in shoots and roots [6], with roots showing higher oxidative stress responses [7] and differential modulation of ROS production and scavenging mechanisms [8] compared to shoots. Roots also depend on the GMF for proper Fe and Cu uptake efficiency and plant mineral nutrition [9,10], as well as correct lipid metabolism [11]. The GMF is also important for the efficiency of the photosynthetic machinery [1] and interacts with plant flowering by modulating flowering-related genes [5], clock gene amplitude [12], and the gene expression pathways downstream of cryptochrome and phytochrome photoactivation [13]. However, plant magnetoreception also occurs in dark conditions [14] and shows light-independent responses [12]. Although the mechanism of plant magnetoreception is far from being clear, several lines of evidence indicate that plant iron-sulfur complex assembly (IscA) scaffold proteins [15] might act as magnetic sensors by forming a magnetosensor (MagS) complex with cryptochrome (Cry), as found in pigeons [16].

Among the several players in plant growth and development, a growing body of evidence suggests an important role of 14-3-3 proteins. 14-3-3s are dimeric 30 kDa proteins widely distributed in eukaryotes, where they are localized in various organelles, including the nucleus, mitochondria and plastids, as well as in the cytosol and the plasma membrane (PM) [17]. In plants, 14-3-3 proteins play a role in carbon and nitrogen metabolism, hormone signaling and biosynthesis, and regulation of ion transport. 

14-3-3 proteins exist as multiple isoforms [18], and their typical trait is the ability to bind target proteins through the recognition of phosphorylated consensus motifs [19,20,21].

According to the biochemical function of the target, 14-3-3 binding can lead to different functional consequences, such as regulation of subcellular localization, enzymatic activity, protein stability, or alteration of protein–protein interactions [17]. 14-3-3 proteins can also participate in different signaling pathways by transmitting signals to specific responsive genes, thus playing a role as molecular switches in coordinating plant responses to both biotic and abiotic stress [22,23]. For instance, RNAi silencing of 14-3-3 proteins increases ROS production, indicating their involvement in plant defense against abiotic stress [24], and has been shown to have a positive correlation with the antioxidant system [25]. On the other hand, the upregulation of 14-3-3 proteins enhances stress tolerance [26] by activating the ROS-scavenging system [27] and increasing the levels of stress-responsive proteins [26]. Moreover, 14-3-3 over-expression increases the gene expression of the ABA signaling pathway and ion transport [27]. 

One of the most important functions of 14-3-3 proteins is the regulation of the PM H^+^-ATPase, which plays a key role in many essential processes, including ion homeostasis, cell expansion, and responses to various stresses [28,29,30]. 14-3-3 proteins regulate the activity of the PM H^+^-ATPase by binding to its C-terminal autoinhibitory domain [31], thereby leading to its displacement and, consequently, to enzyme activation [32]. The H^+^-ATPase binding site for 14-3-3 proteins is generated upon phosphorylation of a conserved Thr residue within the sequence Tyr-Thr-Val, located at the very end of the C-terminus [33]. From a morphological point of view, the interaction of 14-3-3 with H^+^-ATPase stimulates cell elongation and eventually plant growth, as shown in the root apex of maize (*Zea mays*) [34]. Indeed, 14-3-3 proteins participate in regulating primary root growth, root hair development, and geotropic growth [17]. 

Given the involvement of 14-3-3 proteins in stress response and the importance of the PM proton pump in a multitude of cellular processes, in this work we aimed to assess whether 14-3-3 proteins and the PM H^+^-ATPase are involved in plant magnetoreception by responding to GMF variations. To this end, we lowered the GMF to NNMF values and used maize etiolated seedlings to evaluate the interaction between 14-3-3 proteins and H^+^-ATPase, their content and activities and the phosphorylation level. We also studied the maize morphological changes and assessed the expression of several genes known to be regulated by 14-3-3 proteins. Among these, we focused on responses to some phytohormones and ROS, of which we also assessed the quantitative variations. Overall, our results show that 14-3-3 proteins and the PM H^+^-ATPase are involved in plant responses to magnetic induction.

## 2. Results

### 2.1. The GMF Is Required for Maize Root Development

To evaluate the effect of reduced GMF on the growth of maize seedlings, seeds were germinated, and the plantlets were grown hydroponically in the dark under either GMF or NNMF conditions. These studies were carried out with etiolated shoots because a more efficient purification of PM for activity studies is achievable in these conditions [35]. Moreover, plants show a light-independent magnetic induction mechanism [12,14].

The reduced GMF did not alter the germination rate. After 2 days, no significant differences (*p* > 0.05) were found in the germination rate between plants exposed to GMF and NNMF conditions (*n* = 781 for GMF coleoptiles and roots, *n* = 730 for NNMF coleoptiles and roots). After 5 days, the seedlings were harvested (Figure 1A), and the lengths of coleoptiles and roots were assessed. As shown in Figure 1B, the length of the shoot was not affected by lowering the GMF. Intriguingly, root length was instead significantly (*p* < 0.05) reduced (−17.2%) in plants grown in NNMF, compared to control (GMF) plants. These results indicate that in maize, the GMF is required for proper root growth.

### 2.2. The GMF Is Required for Proper Activity of the Plasma Membrane H^+^-ATPase

Given the reduced root growth and considering the crucial role played by H^+^-ATPase in plant growth, we tested whether NNMF conditions resulted in changes in the phosphohydrolytic activity of the proton pump. Hence, we purified the PM fraction by two-phase-partitioning and assayed in vitro the vanadate-sensitive phosphohydrolytic activity. As shown in Figure 2, no significant differences were found in the H^+^-ATPase activity of PMs purified from coleoptiles of plants grown in NNMF or GMF conditions. On the contrary, the activity was strongly inhibited (−51.4%, *p* < 0.01) in PMs purified from roots.

The reduced H^+^-ATPase phosphohydrolytic activity of PMs purified from roots grown in NNMF could be the result of the inhibition of gene expression and/or a post-translational mechanism involving 14-3-3 proteins. We first analyzed by qRT-PCR the expression of the *MHA2* gene, encoding for the major isoform of the maize PM H^+^-ATPase, being highly expressed in all organs [36]. Moreover, it is well known that the MHA2 H^+^-ATPase plays a crucial role in regulating cell expansion and plant growth in maize seedlings [36]. As shown in Figure 3A, no significant (*p* > 0.05) differences in *MHA2* expression were found in both coleoptiles and roots of seedlings grown in either NNMF or GMF conditions. Accordingly, the amount of H^+^-ATPase in PM fractions was not affected by NNMF (Figure 3B).

### 2.3. The GMF Is Required for the Phosphorylation of the 14-3-3 Protein Binding Site and the Binding of 14-3-3 Proteins

The observed decreased PM H^+^-ATPase activity in roots from plants grown in NNMF conditions could be due to a post-translational mechanism involving 14-3-3 proteins. Hence, we first assessed the expression of the two major 14-3-3 isoforms of maize, *GF14-6* (G-box Regulating Factor 6) and *GF14-12*. As shown in Figure 4A,B, the expression of both genes was not affected by NNMF conditions. Accordingly, the amount of 14-3-3 proteins in the cytosolic fraction was not affected by NNMF (Figure 4C).

To ascertain whether the reduced phosphohydrolytic activity was ascribable to a post-translational mechanism that regulates the binding of activatory 14-3-3 proteins on the proton pump, we first analyzed the amount of 14-3-3 proteins associated with the PM fraction. In fact, it is well ascertained that H^+^-ATPase is the main PM target of 14-3-3 proteins, and its activation is strictly related to membrane-bound 14-3-3 levels. Immunodecoration with anti-14-3-3 antibodies revealed a remarkable decrease (−34.8% by densitometric analysis) in 14-3-3 levels in membranes of roots grown in NNMF (Figure 5A). Since this effect was not due to a reduction in overall 14-3-3 concentration (Figure 4C), our results suggest that NNMF triggered the inhibition of H^+^-ATPase activity by lowering the association of stimulatory 14-3-3 proteins. To confirm this hypothesis, an in vitro overlay assay was performed. Plasma membrane samples from coleoptiles and roots grown in either NNMF or GMF were subjected to SDS–PAGE, blotted onto a nitrocellulose membrane, and probed with recombinant 14-3-3 proteins. As shown in Figure 5B, the H^+^-ATPase from NNMF roots had a reduced ability (−38.5%) to interact in vitro with 14-3-3 proteins.

Since it is well established that the association of 14-3-3 proteins to the proton pump requires the phosphorylation of a threonine within the binding sequence Tyr-Thr947-Val, located at the extreme end of the H^+^-ATPase C-terminus [33], it is conceivable that the hampered binding induced by NNMF was due to reduced Thr947 dephosphorylation. Therefore, we tested the Thr phosphorylation status of H^+^-ATPase using anti-phosphoThr antibodies. As shown in Figure 5C, NNMF triggered a strong decrease (−52.0%) in Thr phosphorylation of the proton pump purified from the roots of seedlings grown in NNMF. 

Hence, our overall data show that the inhibition of H^+^-ATPase activity in roots of seedlings grown under NNMF conditions was due to a post-translational mechanism capable of reducing the phosphorylation of the 14-3-3 protein binding site and consequently the binding of 14-3-3 proteins.

### 2.4. The GMF Induces an Oxidative State in Both Maize Coleoptiles and Roots 

14-3-3 proteins also have a role in the regulation of ROS production in response to stress [37]. Hence, the reduction of 14-3-3 binding to the plasma membrane might, in turn, trigger the alteration of the membrane-bound enzymes involved in redox reactions [26]. To assess the oxidative state of both coleoptiles and roots, we first measured the total H_2_O_2_ production in plants exposed to GMF or NNMF.

The reduction of GMF lowered the level of H_2_O_2_. In particular, coleoptiles exposed to NNMF experienced a 2-fold decrease in H_2_O_2_ content, whereas in roots the reduction was about 1.5-fold (Figure 6).

The reduction of H_2_O_2_ levels triggered by NNMF was associated with the modulation of the main genes coding for ROS-producing and scavenging enzymes. In maize roots exposed to NNMF, we found a downregulation of plastidial ascorbate peroxidase1 (*APX1*) and the PM-located NADPH oxidase (also known as the respiratory burst oxidase homolog, RBOH) *RBOH1*. On the other hand, the superoxide dismutase1 (*SOD1*) gene expression was not affected by the GMF reduction (Table 1). In coleoptiles, exposure to NNMF induced the downregulation of catalase1 (*CAT1*) and *APX1*, whereas *RBOH1* and *SOD1* expression was below a 2-fold change in the NNMF/GMF ratio (Table 1). 

Overall, these data indicate that the GMF is required for a proper oxidative state and that the reduction of GMF to NNMF, which we observed to reduce 14-3-3 binding to the PM, also affects both production and expression of genes related to ROS production and/or scavenging.

An important role of melatonin has been associated with its antioxidant properties [38]. When the melatonin content was measured in maize seedlings exposed to NNMF, we found a strong increase (*p* < 0.05) in roots (15.51 ng g^−1^ FW in NNMF vs. 4.08 ng g^−1^ FW in GMF) and a moderate increase in the coleoptiles (3.72 ng g^−1^ FW in NNMF vs. 2.85 ng g^−1^ FW in GMF). These results suggest a still unknown role of melatonin in plant redox homeostasis.

### 2.5. The GMF Is Required for Proper Production and Gene Expression of the Phytohormones Auxin and Cytokinin

Because 14-3-3 proteins are involved in phytohormone signaling, we analyzed the content and expression of genes involved in the metabolism and transport of selected phytohormones. Regarding the phytohormone content, exposure of etiolated maize seedlings to NNMF induced significant (*p* < 0.05) changes in both coleoptiles and roots. Table 2 summarizes the content of phytohormones in maize roots and coleoptiles of seedlings grown under GMF or NNMF conditions.

When the contents of the auxin indole-3-acetic acid (IAA) and the cytokinin *trans*-zeatin (ZEA) were evaluated, we found a reduction (*p* < 0.05) in IAA and a concomitant increase (*p* < 0.05) in ZEA, which led to a statistical (*p* < 0.05) reduction of the IAA/ZEA ratio (Table 2, Figure 7).

To further understand the underlying mechanisms, we then assessed the expression of genes involved in IAA and ZEA metabolism and transport. In roots, we found that NNMF downregulated both *cytokinin oxidase3* (*CKO3*) and the *cytokinin response regulator1* (*CRR1*). Interestingly, GMF reduction did not affect *CKO3* expression. Conversely, an almost 2-fold upregulation was observed for *CRR1* in coleoptiles (Table 3) Moreover, the expression of two genes coding for pin-formed proteins, *PIN1* and *PIN3*, showed no differences in gene expression. However, the *auxin binding protein1* (*ABP1*) was downregulated in roots (Table 3).

### 2.6. Reduction of the GMF Triggers the Modulation of Stress-Related Phytohormones

Among other phytohormones, the content of abscisic acid (ABA), a hormone involved in stress response, was reduced (*p* < 0.05) in coleoptiles, while it increased (*p* < 0.05) in roots upon exposure to NNMF (Table 2). Similarly, jasmonic acid (JA), another stress-related hormone, was always reduced in both organs. On the other hand, the JA precursor, namely 12-oxo-phytodienoic acid (OPDA), as well as the methyl-ester form of JA (methyl jasmonate, MeJA), were reduced only in coleoptiles (Table 2). Finally, salicylic acid (SA) was reduced (*p* < 0.05) only in coleoptiles, whereas its methyl ester (methyl salicylate, MeSA) showed an opposite trend, being increased (*p* < 0.05) in the roots and decreased (*p* < 0.05) in the coleoptiles (Table 2).

We extended our analysis to investigate the expression of hormone-related genes encoding for enzymes or signaling proteins that have been functionally linked with 14-3-3 proteins. We observed that NNMF induced root downregulation of two gibberellin oxidases (*GA20OX3* and *GA2OX1*), with an opposite trend in root and coleoptiles for *GA2OX1* (Table 4). However, we did not observe any significant change in the regulation of *ethylene AP2-EREBP-transcription factor 10* (*EREB10*), *brassinosteroid bri1-like receptor kinase1* (*BRL1*), and *brassinosteroid-signaling kinase1* (*BSK1*) (Table 4).

### 2.7. Gene Expression of Calcium and Potassium Channels Requires the GMF

Because calcium homeostasis has been associated with 14-3-3 proteins and oxidative stress, we evaluated the expression of four genes involved in Ca^2+^ signaling. Maize roots exposed to NNMF showed a downregulation of *calcium-dependent protein kinase1* (*CDPK1*), *calmodulin1* (*CAL1*), *calcium pump1* (*CAP1*), and *calcium exchanger1* (*CAX1*) (Table 5). In the coleoptiles, the expression of the same genes was not regulated by the GMF, except for *CAL1*, which was upregulated (Table 5).

We also assessed the expression of genes coding for cation and anion channels. Exposure to NNMF prompted a downregulation of the root *potassium channel1* (*KCH1*), whereas coleoptiles showed an opposite trend (Table 5). No regulation in both roots and coleoptiles was found for the genes coding for *slow anion channel-associated1* (*SLAC1*) and the aquaporin *plasma membrane intrinsic protein1b* (*PIP1b*) (Table 5).

### 2.8. The GMF Is Necessary for Proper Expression of Some Genes Involved in Primary Metabolism

Root and coleoptile growth depends on primary metabolism availability. Therefore, we assessed the expression of some genes involved in carbohydrate and nitrogen metabolism. When exposed to NNMF, maize seedlings showed an opposite trend in *sucrose synthase1* (*SUS1*) expression, with downregulation in coleoptiles and upregulation in roots (Table 6). *Isocitrate dehydrogenase1* (*IDH1*) and *malate dehydrogenase1* (*MDH1*) were downregulated only in roots, whereas *phosphoenolpyruvate carboxylase1* (*PEP1*) was upregulated both in roots and coleoptiles (Table 6). The gene involved in nitrogen metabolism, *glutamate synthase1* (*GOGAT1*), was downregulated only in maize roots exposed to NNMF (Table 6).

### 2.9. The Maize MagR Homolog ZmISCA2 Is Magnetically Induced along with CRY1

Maize coleoptiles grown in the darkness are not supposed to stimulate photoreceptors. However, reducing the GMF to NNMF always induced coleoptile downregulation of both cryptochromes and phytochromes (Table 7). Interestingly, the MagR homolog *ZmISCA1* showed the same expression pattern of *PHYB1*, whereas *ZmISCA2* gene expression in roots and coleoptiles was the same as that of *CRY1*. Under reduced GMF, *CRY1* and *CRY2* showed an opposite expression trend in roots and coleoptiles (Table 7).

To visualize the differential gene expressions in maize roots and coleoptiles exposed to GMF and NNMF, we produced a cluster heatmap (Figure 8). Gene expression in roots shows the most remarkable difference between NNMF- and GMF-exposed plants, with particular reference to ROS, ions and some genes involved in carbon and nitrogen metabolism. In coleoptiles, differential expression was found for a lower number of genes. As evidenced by the clustering, our results confirmed a clear differential gene expression between roots and shoots, especially when GMF and NNMF were compared (Figure 8).

## 3. Discussion

The GMF is a natural component of the biosphere. During evolution, all organisms experienced the presence of the GMF, and some organisms evolved the ability to use magnetic fields (MF) during migration [39] or for orientation [40]. In plants, the mechanism of magnetoreception has been inferred for both the radical pair mechanism (RPM) of chemical magnetosensing and the MagR/Cry biocompass [15,41]. Both mechanisms entail alterations of the rates of redox reactions and subsequently altered concentrations of free radicals and ROS [8,42,43]. The cascade of events following the perception of MF variations is supposedly wide and implies the activation of sensing molecules (e.g., MagR) and the transduction of the signal involving second messengers, transcription factors, and DEGs [6]. In this work, we showed that 14-3-3 proteins and the PM H^+^-ATPase respond to reduction of the MF and that these events are associated with a general reduction in the oxidative state and modulation of maize phytohormones and ion transport.

### 3.1. GMF Modulates the 14-3-3-Mediated Activation of the Root PM H^+^-ATPase

We have provided evidence of the ability of the GMF to modulate the proton pump activity of maize roots through the regulation of 14-3-3 protein binding on the enzyme. Under NNMF conditions, we observed a drastic reduction in the amounts of 14-3-3 proteins associated with PM, which was correlated with the level of Tyr phosphorylation of H^+^-ATPase and its ability to interact in vitro with 14-3-3 proteins. Hence, this body of evidence demonstrates that GMF influences H^+^-ATPase activity by modulating the phosphorylation status of the conserved Thr within the 14-3-3 consensus motif located at the extreme C terminus of the enzyme. Many signals, including auxin, ABA, blue light, and sucrose, regulate phosphorylation levels of the penultimate Thr in the C-terminus of H^+^-ATPase [44]. In this respect, it is noteworthy that NNMF triggered significant perturbations of hormone homeostasis and alterations of carbon metabolism (see Section 3.2), which might be responsible, at least partially, for the reduced Thr phosphorylation and subsequent 14-3-3 binding.

Besides the functional role of melatonin in vertebrate timekeeping, this compound is widely present in plants, with levels much higher than the known physiological concentrations in the blood of many vertebrates [38]. In plants, melatonin’s important role might be associated with its antioxidant properties [45]. Therefore, increased melatonin content might compensate for the reduction in the major ROS scavenging systems observed under NNMF. Interestingly, the increased melatonin level was associated with 14-3-3 variation in roots but not in coleoptiles, leaving an open question on the role of melatonin/14-3-3 interaction in plant root magnetic induction.

### 3.2. Oxidative State, Second Messengers, and Phytohormone Homeostasis Play a Role in Plant Magnetic Induction

In response to GMF variations, plants activate genes and enzymes involved in ROS signaling pathways [6], with a mechanism that appears to be shared in different organisms [6]. In *Arabidopsis thaliana*, GMF reduction induces metabolomic and transcriptomic modulation of ROS-scavenging enzymes and the regulation of antioxidant polyphenols [8]. In maize roots exposed to NNMF, the reduction in H_2_O_2_ production and the downregulation of *RBOH1* are associated with the downregulation of genes involved in calcium signaling, including *CDPK1*, *CAL1*, *CAP1*, and *CAX1*. RBOHs are regulated by cytosolic calcium and protein phosphorylation [46], and some RBOHs have been found to activate ROS signaling [47]. 14-3-3 proteins can modulate the activity of CDPK signal transduction pathways [48] and might be involved in plant magnetic induction. Moreover, the downregulation of cytokinin oxidases, which are involved in the inactivation of gibberellins [49], and *CKO3*, which is involved in the catabolic deactivation of cytokinins [50], were associated with the increased content of ZEA and the decreased oxidative state under NNMF conditions. 

The general lowering of the maize root oxidative status under NNMF is also correlated to the decrease in *ABP1* gene expression and IAA content, which are fundamental for the proper activity of the proton pump. ABP1 interacts at the surface of the PM directly with ion channels and H^+^-ATPase [51] by increasing the proton pump activity. In NNMF conditions, this eventually reduces the root growth rate. Furthermore, the increased levels of ZEA lower the IAA/ZEA ratio and cause the reduction of root growth. In fact, the coordinated root growth depends on auxin-dependent and cytokinin-dependent programs of gene activity, including the cytokinin response regulator *CRR1* [52]. The root reduction was also associated with increased levels of ABA. Indeed, ABA promotes senescence and regulates root growth and the root/shoot ratio [53]. 

From a carbon and nitrogen metabolic point of view, sucrose loading is stimulated by IAA and inhibited by ABA, whereas sucrose uptake by sinks shows the opposite [54]. Our results suggest that NNMF, by altering the IAA and ABA levels, might influence the gene expression of sucrose synthase. Indeed, *SUS1* expression showed an increase in roots when ABA content was high, whereas its expression was decreased in coleoptiles when ABA concentrations were reduced.

14-3-3 proteins interact with components of the signaling pathways of JA [55] and SA [56]. JA is a modulator of auxin responsiveness [57], and IAA/JA interaction has been found to affect plant root system development [58]. A notable crosstalk exists between SA and ROS. SA, along with MeJA and ABA, is a positive regulator of senescence. MeSA is involved in systemic responses. Moreover, MeJA and MeSA are important components of signal transduction cascades triggered by Ca^2+^ signaling [59], which is affected by NNMF conditions [60].

Overall, these data indicate that maize magnetic induction is triggered by early events that involve calcium signaling, H^+^-ATPase activity, and the binding of 14-3-3 proteins that are associated with the reduction in the growth-promoting phytohormone IAA in the roots and the increase in content of cytokinins in both organs. These events are associated with the increased content of stress-related phytohormone ABA that collectively causes reduced root growth.

### 3.3. Maize Magnetic Induction Is Associated with ZmISCA and Photoreceptor Modulation

Responses of plants to changing MF have been linked to cryptochrome-dependent signaling pathways [61]. Enhancing GMF induces cryptochrome responses to light, but plant responses to varying MF have been observed even exclusively in the dark, indicating that the magnetically sensitive reaction step in the cryptochrome photocycle occurs during flavin reoxidation and involves the formation of ROS [42]. However, recent findings suggest that light-independent magnetic induction in Arabidopsis might be related more to a level-crossing mechanism than a cryptochrome-associated radical-pair mechanism [14].

In animals, there is a conserved structural architecture for the Cry/MagR complex, and in the monarch butterfly the Cry1/MagR complex displays a strong magnetic polarity in response to an external MF [62]. Our results showed a remarkable overlapping of gene expression between *ZmISCA2* and *CRY1*, which supports our hypothesis that some plant ISCA might be involved in magnetic induction in a way similar to that occurring in animals [15]. On the other hand, *ZmISCA1* gene expression in maize seedlings exposed to NNMF showed the same pattern as that of *PHYB1*. In Arabidopsis, phyB plays a role in mediating NNMF-induced flowering delay, and its activation appears to be attenuated by NNMF conditions. In contrast, its activation under GMF conditions depends on the presence of cryptochromes [13]. Overall, our results point to a differential interaction of *Zm*ISCAs and photoreceptors, with *Zm*ISCA2 being more associated with cry1 and *Zm*ISCA1 showing similar patterns of phyB1 variation. Further studies are underway to better assess the role of these two *Zm*ISCAs in the interplay with photoreceptors during magnetic induction.

## 4. Materials and Methods

### 4.1. Plant Material and Growth Conditions

Maize (*Zea mays* L. cv. DKC6980) caryopses from Dekalb, Mestre, Italy, were grown hydroponically in the dark [35] for 5 days in a handcrafted hydroponic system under either NNMF or GMF conditions. Briefly, 10 × 15 cm plastic boxes were used (4 boxes for each experimental condition). About 50 seeds/box were placed on a plastic grid, and the box was filled with water up to the grid. To maintain an adequate humidity level, a wet sheet of filter paper was placed over the seeds for the first 2 days. The temperature was set and maintained at 24 °C for the entire experimental time by air conditioning. All experiments were performed under normal gravity and atmospheric pressure. Overall, the experiment was repeated four times.

### 4.2. Near Null Magnetic Field (NNMF) Generation System and Plant Exposure

The GMF (or local geomagnetic field) values were typical of the Northern Hemisphere at 45°0′59″ N and 7°36′58″ E coordinates. A near-null magnetic field (NNMF) was generated as previously described [5]. Real-time monitoring of the MF in the plant exposure chamber was achieved with a three-axis magnetic field sensor (model Mag-03, Bartington Instruments, Oxford, UK) as previously described [8]. The current applied through each of the three Helmholtz coil pairs maintained the MF constant inside the plant growth chamber at NNMF intensity, as recently reported [63]. Plants were placed in the geometric center of the triaxial Helmholtz coil system and exposed to NNMF (~40 nT). After the exposure period, coleoptiles and roots were separately harvested, photographed, and immediately frozen in liquid nitrogen.

### 4.3. Analysis of Shoot and Root Growth

The length of shoots and roots of 5-day-old seedlings was measured by analyzing respectively 781 (GMF) and 730 (NNMF) seedlings from four independent experiments, photographed with a digital camera and analyzed with ImageJ software [64].

### 4.4. Plasma Membrane Purification

Plasma membrane samples were purified from etiolated maize coleoptiles and roots according to Serrano [65], with minor modifications [66]. Briefly, 25 g of plant material was homogenized with 25 mL of a buffer containing 25 mM MOPS-BTP, 250 mM sucrose, 2 mM DTT, 5 mM EDTA, 1 mM PMSF, 0.2% BSA, pH 7.8, filtered with a fine weave gauze and centrifuged for 20 min at 8000 g at 4 °C. The supernatant was ultracentrifuged at 70,000× *g* for 30 min, and the pelleted microsomal fraction was resuspended in 2 mL of 5 mM potassium phosphate buffer containing 0.2 mM PMSF, pH 7.8. Plasma membranes were isolated from microsomes by two-phase partitioning. Briefly, 2 mL microsome suspension was added to 14 mL of 5 mM potassium phosphate buffer containing 7.2% Dextran T-500, 7.2% PEG-3350, 286 mM sucrose, 5.7 mM KCl, pH 7.8. The samples were then gently mixed through repeated inversions and centrifuged at 2000× *g* for 15 min at 4 °C. The upper phase was recovered, diluted twofold in 10 mM MOPS-BTP, 250 mM sucrose, 2mM EDTA, 1mM DTT, 1mM PMSF, pH 7.0, and centrifuged for 45 min at 125,000 g at 4 °C. Finally, the pellet was resuspended in 2 mL of 10 mM Tris-HCl, 1 mM EDTA, 1 mM DTT, 20% glycerol, pH 7.6.

### 4.5. H^+^-ATPase Activity

The phosphohydrolytic activity of PM preparations from maize shoots and roots was assayed according to Serrano [65]. In particular, 30 μg of two-phase partitioned PM vesicles was incubated in 500 μL of reaction buffer (50 mM Tris-MES, 5 mM KNO_3_, 5 mM MgSO_4_, 0.2 mM (NH_4_)_6_Mo_7_O_24_, 2 mM ATP, pH 6.5). After 30 min, the reaction was stopped with 1 mL of phosphate reagent (0.5% SDS, 0.5% (NH_4_)_6_Mo_7_O_24_, 2% (*v*/*v*) H_2_SO_4_, 0.01% ascorbate). The phosphomolybdic acid produced during the reaction was determined by measuring the absorbance at 740 nm. The phosphate concentration was then calculated by interpolating with a calibration curve obtained with different concentrations of KH_2_PO_4_. For each sample, the residual activity in the presence of 0.2 mM of the H^+^-ATPase inhibitor orthovanadate was subtracted from the obtained values to calculate the H^+^-ATPase-specific activity.

### 4.6. Immunoblotting

Immunoblotting analysis was performed according to Muzi et al. [67], using polyclonal rabbit anti-14-3-3 antibodies (1:1000), recognizing all plant 14-3-3 isoforms, polyclonal anti-H^+^-ATPase antibodies (1:1000), directed against a conserved region in the C-terminal domain, or polyclonal rabbit anti-phospho-Thr antibodies (Agrisera, Vännäs, Sweden). Immunodecoration was performed using an HRP-conjugated anti-rabbit secondary antibody, as already described [68].

### 4.7. Overlay Assay

The cDNA of the maize 14-3-3 isoform GF14-6 (G-BOX FACTOR14-6) was cloned into the pGEX-2T vector, expressed in *Escherichia coli* as a GST-fused protein [35]. The GST-14-3-3 was immobilized onto a Glutathione Sepharose 4B resin (Merck KGaA, Darmstadt, Germany), and the 14-3-3 protein eluted by thrombin cleavage as already described [69]. The overlay assay was performed according to Visconti et al. [70], with slight modifications. Ten micrograms of two-phase partitioned plasma membrane material was subjected to SDS-PAGE and blotted onto a PVDF membrane. The membrane was blocked with 5% fatty-acid-free milk in buffer H (25 mM Hepes-OH, 75 mM KCl, 5 mM MgCl_2_, 0.1 mM EDTA, 1 mM DTT, 0.04% Tween-20, pH 7.5) and then incubated overnight at 4 °C in buffer H containing 3% fatty-acid-free milk and 50 µg/mL 14-3-3. The membrane was washed extensively with buffer H and incubated with the anti-GF14-6 antibody (1:600) in the same buffer. Immunodecoration was performed as indicated in Section 4.6. The densitometric analysis was performed using ImageJ image-processing software [64]. The densitometric values are expressed as a percentage of the maximum integrated densitometric value (the product of the area and mean grey value).

### 4.8. RNA Extraction and Quantitative Gene Expression

The total RNA was extracted from maize coleoptiles and roots using peqGOLD TriFast (VWR International, Radnor, PA, USA) according to the manufacturer’s instructions. The RNA extracted was quantified employing BioSpec-Nano (Shimadzu, Kyoto, Japan). One µg of total RNA from each sample was reverse transcribed into double-stranded cDNA using the High Capacity cDNA Reverse Transcription Kit (Thermo-Fisher Scientific, Waltham, USA) following the manufacturer’s recommendations. Gene expression analysis was performed on cDNA obtained by real-time qPCR with a QuantStudio 3 Real-Time PCR System (Applied Biosystem, Waltham, MA, USA), and the reactions were performed by employing PerfeCTa SYBR Green FastMix, ROX (Quantabio, Beverly, MA, USA) in 15µL of total volume. The thermocycling was performed using a two-step cycling protocol: 50 °C for 2 min, 95 °C for 10 min, 95 °C for 15 s, and 60 °C for 60 s. The last two steps were repeated for 40 cycles. All primers were designed using Primer 3 software. Four different reference genes, *ACT1* (Zm00001d010159), *GAPC2* (Zm00001d035156), *TUB1* (Zm00001eb000490), and *ELFA2* (Zm00001eb285220), were used to normalize the results of qPCR; the most stable gene was *ACT1*. All amplification plots were analyzed using QuantStudio Design and Analysis software (Applied Biosystem, Foster City, CA, USA) to obtain Ct values. Relative RNA levels were calibrated and normalized with the level of ACT1 mRNA. The primers list used for the amplification is available in Appendix A.

### 4.9. Protein and ROS Quantification

The protein concentration was determined by the method of Bradford [71], using bovine serum albumin as the standard.

The total hydrogen peroxide content was measured with the MAK311-1KT Peroxide Assay Kit (Sigma-Aldrich, St. Louis, MO, USA) according to the manufacturer’s instructions. Around 70 mg of frozen maize ground tissue were extracted in 1:10 (*w*/*v*) mQ water. Samples were quickly vortexed and centrifuged at 15,000× *g* for 10 min. Samples were assessed immediately by transferring 40 µL of the supernatant into a 96-well flat-bottom plate. After incubation for 30 min at room temperature with 200 µL of assay kit Detection Reagent, the absorbance was measured at 600 nm with an NB-12-0035 Microplate Reader (NeoBiotech, Nanterre, FR). The H_2_O_2_ content was calculated from a standard curve obtained by dilutions of 3% hydrogen peroxide solution. H_2_O_2_ quantification was performed on the same material used for gene expression analysis.

### 4.10. Phytohormone and Melatonin Analysis

The phytohormone content was determined following a previously reported protocol [72]. Briefly, three biological replicates of maize coleoptiles and roots (about 100 mg) from plants grown under GMF or NNMF conditions were extracted with 1 mL of 1:1 (*v*/*v*) Methyl-t-Butyl Ether (MTBE)/MeOH. After mixing by vortex, the samples were cold sonicated for 15 min. After sonication, 0.5 mL of 0.1% HCl acidified water was added. After 5 min at 4 °C, samples were centrifuged at 10,000× *g* for 10 min at 4 °C, and the upper phase consisting of MTBE was transferred to a clean tube and dried under a nitrogen flow. The dried pellets were resuspended in 100 μL 50% (*v*/*v*) propanol solution. An Agilent Technologies (Santa Clara, CA, USA) High-Performance Liquid Chromatography (HPLC) system, with modules consisting of an auto-injector, a column oven, two binary pumps, a degasser, and a diode array detector (DAD), was used for separation. Analytical separation was achieved by flushing a binary solvent system consisting of water acidified with 0.1% (*v*/*v*) formic acid (solvent A) and methanol acidified to 0.1% (*v*/*v*) with formic acid (solvent B) and using a C18 Luna reverse phase column (3.00 μm, 150 × 3.0 mm i.d.) (Phenomenex, Aschaffenburg, Germany). The gradient was as follows: 0–8 min 95% Solvent A; 8-12 min Solvent A decreased to 75%; 12–16 min Eluent A decreased to 55%; 16–32 min Solvent A decreased to 25%; and after that, it was reduced to 5% at 45 min. Solvent A was maintained at this concentration for 10 min. Before the next injection, chromatographic conditions were restored to initial conditions and maintained for 8 min. During the chromatographic run, the flow rate was set to 0.2 mL min^−1^, while the injection volume was 10 μL. Identification of phytohormones was achieved by tandem mass spectrometry (MS/MS), performed using an ion trap (Bruker Daltonics, Billerica, MA, USA) in multiple reaction monitoring (MRM) scan type and with an electrospray ionization (ESI) source. MRM transitions for pure standards and analytes were obtained by setting the ion polarity to either positive (for IAA, OPDA, ZEA, and Melatonin) or negative (JA, SA, ABA) modality, with a spray voltage at 4.5 kV for both ion polarities. Due to the low ionization and fragmentation of the methylated forms of JA and SA, MeJA and MeSA were detected and quantified using DAD set at 260 nm. For MS/MS analysis, the source temperature was set to 500 °C, while nitrogen was flushed at 25 psi. Quantification of phytohormones was performed using external calibration standards. Standard mixtures containing all phytohormones analyzed were prepared from the stock solution of individual phytohormones by dilution in 50 percent (*v*/*v*) propanol at six different concentrations (1000, 100, 10, 1, 0.1, and 0.01 ng/mL). Calibration curves were performed by plotting the peak areas of the hormones against their concentrations, and the curves were fitted by linear regression. The limit of detection (LOD) was defined as a signal (S)/background noise (N) ratio of S/N > 3, while the limit of quantification (LOQ) was defined by S/N > 10, in accordance with [72].

### 4.11. Statistical Analysis

The data were treated by using Systat 10. Mean value was calculated along with the SD. Paired *t*-test and Bonferroni-adjusted probability were used to assess the difference between treatments and controls. A heatmap was obtained with Heatmapper (http://www.heatmapper.ca/, accessed on 4 July 2023) [73] by using Pearson clustering with the single linkage method. All values are expressed as means ± standard deviation. Data on ROS and hormones are expressed on a dry weight basis.

## 5. Conclusions

In this work, we reported for the first time the effect of MF reduction on 14-3-3 proteins and the PM proton pump H^+^-ATPase, and we showed that NNMF conditions alter the proton pump activity by reducing the binding of 14-3-3 proteins. It is known that at weak MFs in the range 40–35,000 nT, different biological effects can be observed. Below the GMF values, the effects of a hypomagnetic field (HMF) occupy a special place [74]. Besides MagR and its interaction with cryptochrome, molecular or biophysical targets for magnetoreception are protein complexes that interact with the GMF. Rotations of molecules, as happens in the mitochondrial and chloroplast ATP synthase and the vacuolar H^+^-ATPase (V-ATPase), can significantly affect the process in which the magnetic moment initiates subsequent biophysical events and pose important questions regarding the role of molecular rotations in magnetic biology [74]. 14-3-3 proteins specifically interact with the catalytic β-subunit of the ATP synthase [75] and the A-subunit of the V-ATPase [76]. Here, we showed that the PM H^+^-ATPase and its association with 14-3-3 proteins are potential targets of maize magnetic induction, with roots playing a major role in maize magnetoreception. Alteration of the 14-3-3 binding to the PM is associated with a general reduction of the oxidation state, with a significant reduction of both H_2_O_2_ and downregulation of genes coding for enzymes involved in ROS production and scavenging and calcium homeostasis. Recently, activation of H^+^-ATPase by the increased calcium influx has been found to be involved in magnetic induction [77]. These early events are followed by the downregulation of IAA synthesis and gene expression and the increases in both cytokinin and ABA, which cause a reduction in root growth. Our results confirm that, as shown in Lima bean [1] and Arabidopsis [15], ISCAs are modulated by the reduction of the GMF in maize as well, and we showed that the MagR homolog, *ZmISCA2*, displays a pattern of expression similar to that of *CRY1*.

Nevertheless, it is important to consider that the observed effects on H^+^-ATPases, 14-3-3 proteins, phosphorylation and gene expression might also be correlated to differences in developmental stage and/or growth speed that are also known to be the result of GMF variations [5]. Further studies will evaluate the causal relationship between GMF reduction and the differential growth responses of maize seedlings in development and/or speed of growth.

MF variations are perceived by plants as an abiotic stress, with the modulation of several stress-related phytohormones. Interestingly, melatonin, a widespread molecule present in many kingdoms, is increased by GMF reduction. This event can be interpreted as a second line of defense against oxidative stress, due to melatonin antioxidant properties [47] or, more intriguingly, to a still unknown role of melatonin in magnetic induction.

## Figures and Tables

**Figure 1 plants-12-02887-f001:**
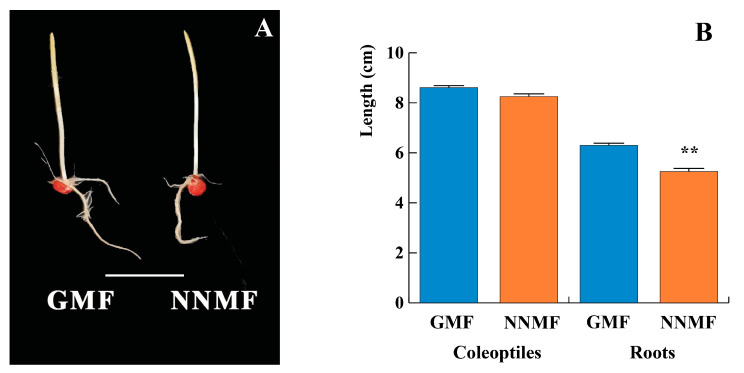
Maize seedlings exposed to either geomagnetic field (GMF) or near-null magnetic field (NNMF) conditions. Panel (**A**), representative seedlings collected after 5 days of growth in the darkness. Metric bar 40 mm. Panel (**B**), morphometric analysis of both coleoptiles and roots of GMF and NNMF exposed seedlings. Data given are the mean of four independent experiments. Statistical significance was assessed by unpaired Student’s *t*-test. Metric bars indicate standard deviation, and asterisks indicate highly significant differences (**, *p* < 0.01) between GMF and NNMF.

**Figure 2 plants-12-02887-f002:**
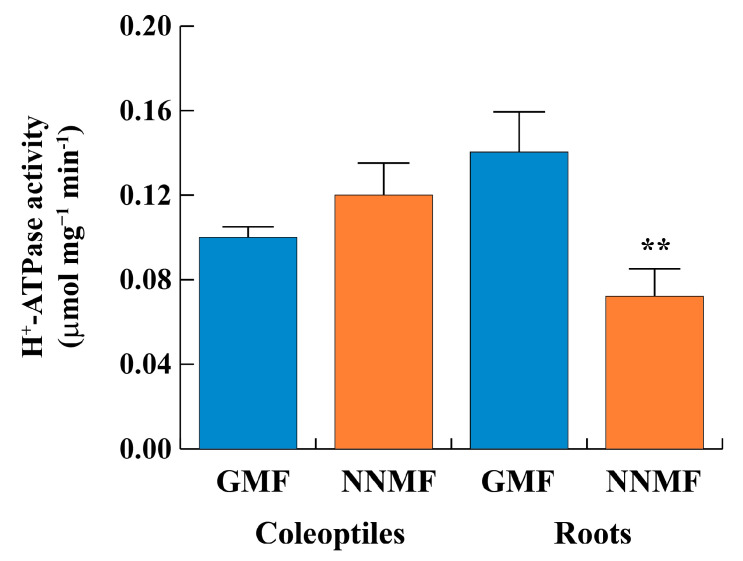
Vanadate-sensitive phosphohydrolytic activity of H^+^-ATPase from plasma membrane purified fractions of maize roots and coleoptiles exposed to either GMF or NNMF conditions. Data given are the mean of four independent experiments. Statistical significance was assessed by unpaired Student’s *t*-test. Metric bars indicate standard deviation, while asterisks indicate highly significant differences (**, *p* < 0.01) between GMF and NNMF.

**Figure 3 plants-12-02887-f003:**
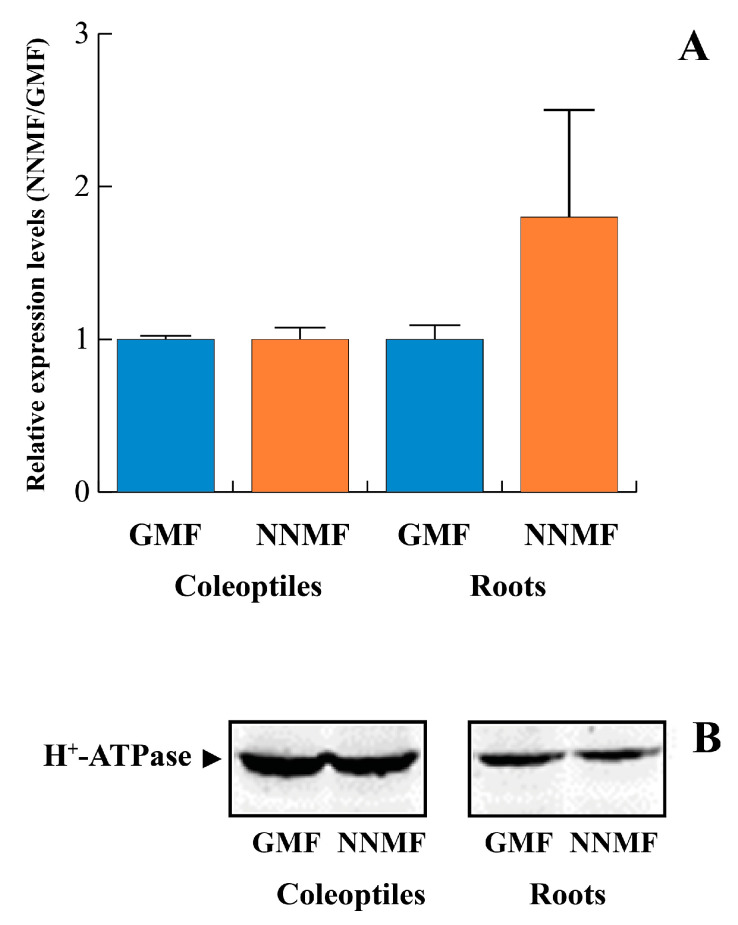
*MHA2* expression and content in maize seedlings exposed to either GMF or NNMF conditions. Panel (**A**), qRT-PCR expression of the *MHA2* gene shows no significant (*p* > 0.05) difference between GMF and NNMF coleoptiles and roots. Data given are the mean of four independent experiments. Statistical significance was assessed by unpaired Student’s *t*-test. Metric bars indicate standard deviation. Panel (**B**), Western blot analysis of PM proteins from GMF and NNMF coleoptiles and roots. Twenty μg of PM proteins was separated by SDS-PAGE, blotted onto PVDF membrane and immunodecorated with polyclonal anti-H^+^-ATPase antibodies.

**Figure 4 plants-12-02887-f004:**
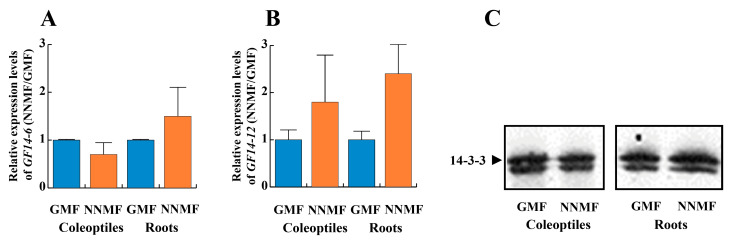
Expression and content of the two major 14-3-3 isoforms of maize, *GF14*-6 and *GF14-12*, in seedlings exposed to either GMF or NNMF. Panel (**A**), qRT-PCR relative expression of *GF14-6*. Panel (**B**), qRT-PCR relative expression of *GF14-12*. Data given are the mean of four independent experiments. Statistical significance was assessed by unpaired Student’s *t*-test. Metric bars indicate standard deviation. Panel (**C**), Western blot analysis of cytosolic proteins from GMF and NNMF coleoptiles and roots. Twenty μg of cytosolic proteins were separated by SDS-PAGE, blotted onto PVDF membrane and immunodecorated with polyclonal anti-14-3-3 antibodies.

**Figure 5 plants-12-02887-f005:**
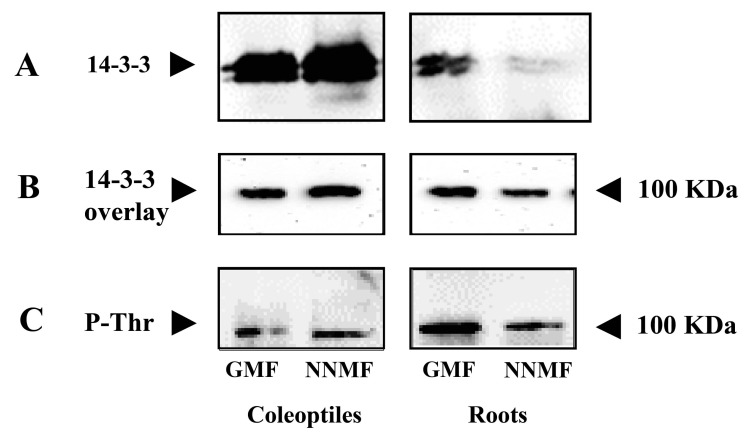
Analysis of the interaction between 14-3-3 proteins and H^+^-ATPase in maize seedlings exposed to either GMF or NNMF conditions. Panel (**A**), levels of membrane-bound 14-3-3 proteins. Western blot analysis of PM proteins from GMF and NNMF coleoptiles and roots. Twenty μg of PM proteins were separated by SDS-PAGE, blotted onto PVDF membrane and immunodecorated with polyclonal anti-14-3-3 antibodies. Panel (**B**), in vitro interaction between recombinant 14-3-3 proteins and H^+^-ATPase. Twenty µg plasma membrane were subjected to SDS-PAGE, blotted onto a PVDF membrane and incubated with 50 µg/mL of recombinant maize GF14-6 14-3-3 isoform. After extensive washings, the membrane was immunodecorated with polyclonal anti-14-3-3 antibodies. (**C**), Thr phosphorylation status of H^+^-ATPase. Western blot analysis of PM proteins from GMF and NNMF coleoptiles and roots. Twenty μg of PM proteins were separated by SDS-PAGE, blotted onto PVDF membrane and immunodecorated with polyclonal anti-PhosphoThr antibodies.

**Figure 6 plants-12-02887-f006:**
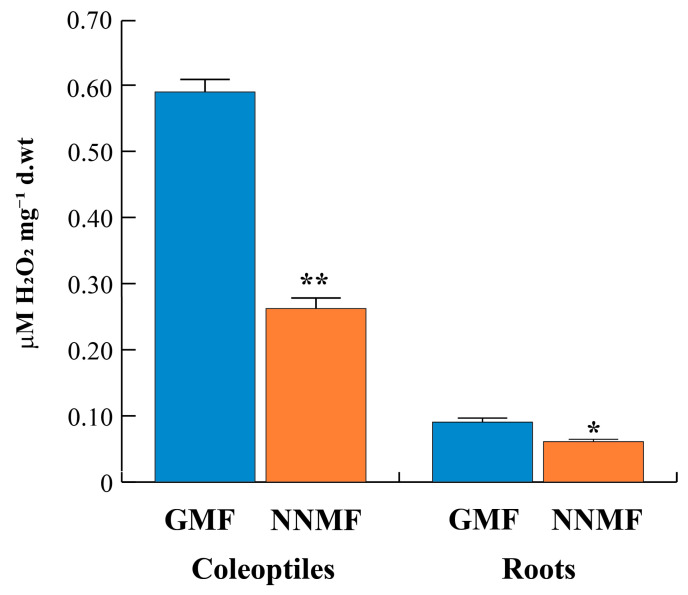
Alteration of the oxidative stress in maize seedlings exposed to reduced GMF conditions. Quantitative determination of hydrogen peroxide (H_2_O_2_) in coleoptiles and roots from plants grown under GMF or NNMF conditions. Data are expressed as µM of H_2_O_2_ mg^−1^ of dry weight (d.wt). Asterisks indicate significant (*, *p* < 0.05) and highly significant (**, *p* < 0.01) differences between GMF and NNMF. Error bars indicate standard deviation.

**Figure 7 plants-12-02887-f007:**
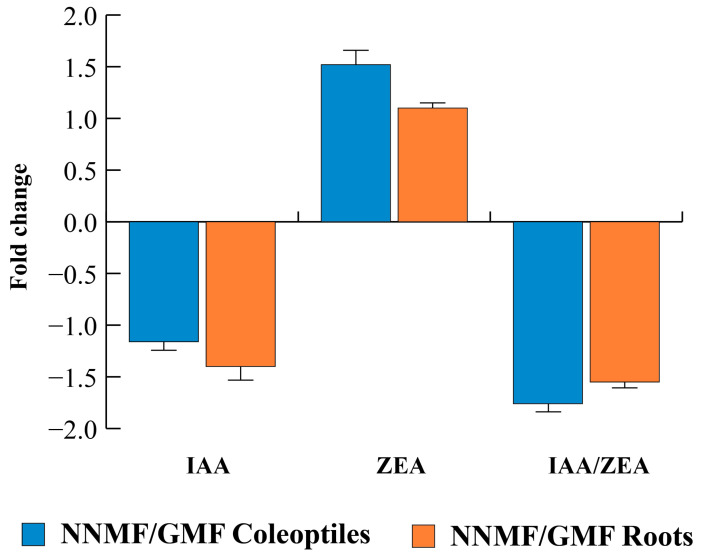
Fold change expressed as NNMF/GMF ratio obtained from the quantitative data reported in Table 2. Error bars indicate standard deviation.

**Figure 8 plants-12-02887-f008:**
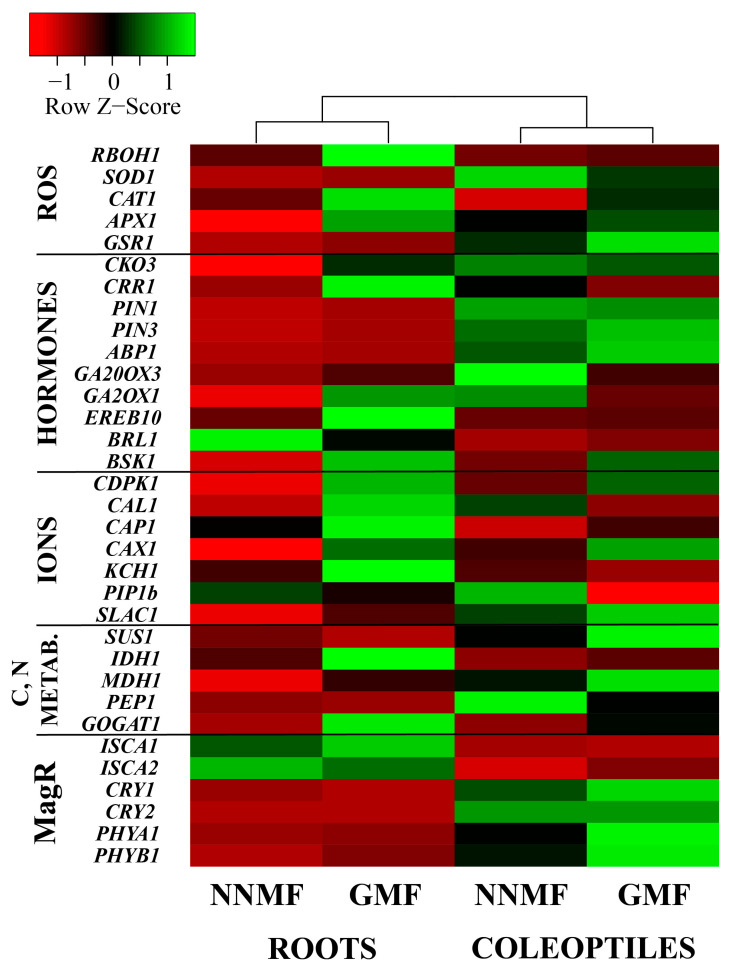
Cluster heatmap of all genes analyzed in this work (Table 3, Table 4, Table 5, Table 6 and Table 7) by considering roots and coleoptiles of NNMF and GMF exposed maize seedlings. The cluster was obtained by using, as clustering method, the single linkage and Pearson distance measuring method.

**Table 1 plants-12-02887-t001:** qRT-PCR analysis of gene expression in coleoptiles and roots exposed to GMF or NNMF. Values are expressed as fold change obtained as NNMF/GMF ratio ± standard deviation. See text for abbreviations.

Gene	Coleoptiles	Roots
*RBOH1*	0.83 ± 0.09	0.22 ± 0.02
*SOD1*	1.36 ± 0.12	0.79 ± 0.08
*CAT1*	0.46 ± 0.09	0.55 ± 0.08
*APX1*	0.29 ± 0.08	0.42 ± 0.06
*GSR1*	0.67 ± 0.08	0.53 ± 0.05

**Table 2 plants-12-02887-t002:** Quantification of phytohormones in coleoptiles and roots of maize seedlings grown under GMF or NNMF conditions. Data are expressed as ng g^−1^ on a dry weight (d.wt) basis. Values are reported as mean values ± standard deviation of three different biological replicates. Different lowercase letters indicate statistical differences as measured by one-way ANOVA followed by Tukey post hoc test at *p* < 0.05. See text for phytohormone abbreviations.

	GMF		NNMF	
	Coleoptiles	Roots	Coleoptiles	Roots
**IAA**	1.793 ± 0.105 ^a^	1.306 ± 0.042 ^b^	1.269 ± 0.041 ^b^	1.135 ± 0.075 ^b^
**ZEA**	24.926 ± 1.729 ^c^	21.916 ± 1.338 ^bc^	27.355 ± 1.888 ^b^	33.675 ± 2.67 ^a^
**ABA**	8.101 ± 0.421 ^c^	11.809 ± 1.049 ^b^	5.068 ± 0.734 ^d^	19.513 ± 1.124 ^a^
**OPDA**	364.44 ± 21.803 ^a^	85.567 ± 6.509 ^c^	315.183 ± 13.064 ^b^	70.691 ± 2.151 ^c^
**JA**	181.856 ± 13.554 ^a^	41.338 ± 3.051 ^c^	154.197 ± 9.129 ^b^	34.767 ± 2.313 ^c^
**MeJA**	13.691 ± 0.685 ^a^	13.126 ± 0.385 ^a^	10.514 ± 0.598 ^b^	13.145 ± 0.648 ^a^
**SA**	143.533 ± 3.902 ^a^	86.798 ± 2.88 ^c^	124.669 ± 8.216 ^b^	85.21 ± 1.697 ^c^
**MeSA**	36.097 ± 1.36 ^a^	6.841 ± 0.149 ^d^	32.413 ± 1.416 ^b^	11.145 ± 0.767 ^c^

**Table 3 plants-12-02887-t003:** qRT-PCR analysis of the expression of some genes involved in the metabolism or transport of IAA and ZEA. Values are expressed as fold change calculated as NNMF/GMF ratio ± standard deviation. See text for abbreviations.

Gene	Coleoptiles	Roots
*CKO3*	0.89 ± 0.19	0.41 ± 0.07
*CRR1*	1.77 ± 0.32	0.23 ± 0.03
*PIN1*	1.17 ± 0.19	0.57 ± 0.03
*PIN3*	0.75 ± 0.07	0.65 ± 0.06
*ABP1*	0.71 ± 0.02	0.46 ± 0.04

**Table 4 plants-12-02887-t004:** qRT-PCR analysis of the expression of hormone-related genes in maize seedlings exposed to GMF or NNMF. Values are expressed as fold change calculated as NNMF/GMF ratio ± standard deviation. See text for abbreviations.

Gene	Coleoptiles	Roots
*GA20OX3*	3.82 ± 0.62	0.43 ± 0.05
*GA2OX1*	1.37 ± 0.38	0.64 ± 0.09
*EREB10*	1.17 ± 0.18	0.57 ± 0.06
*BRL1*	0.41 ± 0.11	2.22 ± 0.12
*BSK1*	0.83 ± 0.08	0.61 ± 0.06

**Table 5 plants-12-02887-t005:** qRT-PCR analysis of the expression of genes coding for proteins involved in Ca^2+^ signaling and ion and water channels in maize seedlings exposed to GMF or NNMF. Values are expressed as fold change calculated as NNMF/GMF ratio ± standard deviation. See text for abbreviations.

Gene	Coleoptiles	Roots
*CDPK1*	1.08 ± 0.25	0.49 ± 0.02
*CAL1*	2.01 ± 0.14	0.24 ± 0.07
*CAP1*	0.62 ± 0.04	0.52 ± 0.06
*CAX1*	0.65 ± 0.03	0.51 ± 0.06
*KCH1*	2.21 ± 0.32	0.28 ± 0.01
*PIP1b*	1.46 ± 0.09	1.03 ± 0.09
*SLAC1*	0.94 ± 0.22	0.61 ± 0.05

**Table 6 plants-12-02887-t006:** qRT-PCR expression analysis of genes coding for enzymes involved in carbohydrate and nitrogen metabolism in maize seedlings exposed to GMF or NNMF. Values are expressed as fold change calculated as NNMF/GMF ratio ± standard deviation. See text for abbreviations.

Gene	Coleoptiles	Roots
*SUS1*	0.53 ± 0.08	1.97 ± 0.21
*IDH1*	0.88 ± 0.11	0.38 ± 0.07
*MDH1*	0.81 ± 0.13	0.53 ± 0.03
*PEP1*	3.14 ± 0.38	2.13 ± 0.33
*GOGAT1*	0.67 ± 0.04	0.41 ± 0.02

**Table 7 plants-12-02887-t007:** qRT-PCR expression analysis of genes coding for two MagR homologs and some photoreceptors in maize seedlings exposed in the dark to GMF or NNMF. Values are expressed as fold change calculated as NNMF/GMF ratio ± standard deviation. See text for abbreviations.

Gene	Coleoptiles	Roots
*ZmISCA1*	0.83 ± 0.07	0.84 ± 0.11
*ZmISCA2*	0.79 ± 0.11	1.21 ± 0.11
*CRY1*	0.76 ± 0.17	1.28 ± 0.16
*CRY2*	1.21 ± 0.16	0.98 ± 0.07
*PHYA1*	0.63 ± 0.11	0.99 ± 0.06
*PHYB1*	0.63 ± 0.04	0.64 ± 0.07

## Data Availability

Data are available upon request.

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
