# Peer review of "14-3-3 Proteins and the Plasma Membrane H+-ATPase Are Involved in Maize (Zea mays) Magnetic Induction"

_plants, 2023, doi:10.3390/plants12152887_

Round 1
Reviewer 1 Report
Thank you for giving me a review of the manuscript written by Anna Fiorillo et al. This is an extremely interesting topic. This is the first time I come across research on the influence of the magnetic field on the activity of the plasma membrane proton pump. Admittedly, on our Earth, we are not in danger of a situation where there will be no magnetic field, at least in the context of the next hundreds of years. As far as I know, it has only decreased by about 10% in the last 150 years. These studies are therefore more in the context of supplementing knowledge. Or they may be important in the possible consideration of plant breeding in the future in space.
Nevertheless, I think it is a very well-written manuscript. The research is conducted reliably and well described. The article consists of a very large number of performed analyses. However, they all make sense to explain how the activity of the plasma membrane proton pump is inhibited in an environment devoid of a magnetic field. I am impressed by the amount of work that went into this manuscript.
In my opinion, the manuscript is suitable for publication as is.
Author Response
We thank very much the reviewer for the appreciation of our work.
Reviewer 2 Report
This manuscript presents results aiming at unravelling the working mechanism of NNMF and MF on maize seedlings with specific attention for H+-ATPASEs and 14-3-3 proteins.
In general the approach idea is good and the assays used seem to be performed well. However, there are some concerns with regard to the basic experimental set up and lack of good description of the procedures and statistical validity. This all related to the production of the plant material which is the key factor determining all further results, analysis and conclusions. To be convinced of all the analysis results the plant material production must be set up without flaws and in a statistically sound way. I cannot find enough information in the manuscript to be convinced of this. In addition, some additional data may be required.
The authors should address the following issues:
1. It is not clear from the text how many seeds were used to produce the seedlings and how many time the experiment (control versus NNMF) was repeated. Based on the statistics for seed germination and seedling tests at least 400 seeds per condition must be used (e.g. see ISTA seed testing procedures www.seedtest.org), in e.g. 4 separate experiments of 100 seeds each per condition.
2. The set up of the seedling growing is not clear although there is some reference to Camoni et al. However, this paper has no information on the method used and refers to another paper, which also refers to another paper (Marra et al., 1992, Plant Phys). In the Marra paper there is also limited information. Specially for the research in the current manuscript to seedling growth set up is crucial and needs attention and detailed description.
3. The set up for the growth of control seedlings is not clear from the manuscript: are they in the same NNMF generation system, but with zero current? As the germination and seedling growth are very sensitive for condition changes it is absolute necessary that the only change in condition is the MF, this also counts for the location in the room, even more as it is not a completely environment controlled germination cabinet. The authors should address this issue.
4. Important for the interpretation of the results is to establish if at NNMF there is a delay in germination (data not shown in the manuscript, but would be good to show it), a change in germination percentage, a delay in growth or a change in morphology. For this we need to follow the growth of the germination and seedling growth in time.
5. Especially as the authors focus on H+-ATPases as target it would be good to examine the cells in the roots for degree of stretching (cell length) to find out if the reduced length is due to less stretching or lower amount of cells.
6. It would be good to also measure the dry-weight of the seedlings (coleoptiles and roots separately) to explore the effect on the dry weight.
7. With regard to the further experiments on the seedling materials it is clear how much plant materials is used (in grams), but not from how many seedlings this was taken as well as from how many treatments and if there was selection of seedlings on size for the samples taken?
8. With regard to the issue of possible delayed growth a lot could be learned from taking seedlings of similar size from both conditions (control and NNMF) to do the follow up tests (most importantly the H+-ATPase activity and expression level test). E.g. do NNMF seedlings with similar size as control (earlier harvested) show the same or different H+ATPAS activity?
9. I expect to see also a MF dose-response curve for seedling growth.
10. Why refer the activity and expression measurements to the fresh weight? Is reference to the dry weight not more logical?
11. It is confusing to show effects smaller than 1 (or -1) as 1 or -1.
12. The discussion is a bit too long. Could be reduced by 30%.
Minor issues:
Lines 132-133: could be result of both (and/or instead of or).
Figures 4 A and B: show on the y-axis what is shown (e.g. GF14-6 etc).
Line 154: “As a result ….” This is not per se a result
Line 154: “of 14-3-3 in “ should be “of 14-3-3 proteins in “
Line 221: “gene expression of ROS”. What is that??
Line 347: “if both ….” Should be “of both …”
Should be corrected for some typos and in some places could be more concise
Author Response
We thank the reviewer for raising some important points that allowed us to describe better the work done and overall improved the quality of our paper. Below are point-to-point answers to the reviewer's comments.
- It is not clear from the text how many seeds were used to produce the seedlings and how many time the experiment (control versus NNMF) was repeated. Based on the statistics for seed germination and seedling tests at least 400 seeds per condition must be used (e.g. see ISTA seed testing procedures www.seedtest.org), in e.g. 4 separate experiments of 100 seeds each per condition.
R: we thank the reviewer for this observation. Indeed, data are the mean of four independent experiments, with more than 700 seedlings analyzed. We better clarified the experimental procedure in the manuscript, also indicating in detail the statistical analysis performed.
- The set up of the seedling growing is not clear although there is some reference to Camoni et al. However, this paper has no information on the method used and refers to another paper, which also refers to another paper (Marra et al., 1992, Plant Phys). In the Marra paper there is also limited information. Specially for the research in the current manuscript to seedling growth set up is crucial and needs attention and detailed description.
R: we are sorry for this lack of accuracy in describing the experiment as was done. Most of the time the editorial board request to cut descriptions present in other papers, but in this case, as the reviewer pointed out, it is important to fully describe the procedure. The revised text now reports the detailed materials and methods used for seedling growth (see 4.1).
- The set up for the growth of control seedlings is not clear from the manuscript: are they in the same NNMF generation system, but with zero current? As the germination and seedling growth are very sensitive for condition changes it is absolute necessary that the only change in condition is the MF, this also counts for the location in the room, even more as it is not a completely environment controlled germination cabinet. The authors should address this issue.
R: as we described in several other papers, our system is built in a lab that has been specifically designed for these experiments. Plants are grown in controlled environment conditions and at the same time plants of controls (without MF reduction) and treatments (in NNMF conditions) are grown. In fact, the GMF changes continuously and in previous experiments we found differences in controls grown in different times of the year. Therefore, we expose simultaneously plant to GMF and NNMF using the same environmental (light, temperature and humidity) conditions.
- Important for the interpretation of the results is to establish if at NNMF there is a delay in germination (data not shown in the manuscript, but would be good to show it), a change in germination percentage, a delay in growth or a change in morphology. For this we need to follow the growth of the germination and seedling growth in time.
R: we thank the reviewer for this observation. In the short period of exposure (2 days) we found no differences in the germination, whereas the changes in morphology have been described in the text. A new sentence in the text explain this important issue. Germination kinetics was not the scope of this work, and we thank the reviewer for this suggestion that will be surely taken in consideration in future works.
- Especially as the authors focus on H+-ATPases as target it would be good to examine the cells in the roots for degree of stretching (cell length) to find out if the reduced length is due to less stretching or lower amount of cells.
R: we thank the reviewer for this interesting remark. Cell biology was not the target of this manuscript. Also the ultrastructure and the cytology of cells were not investigated as they were out of the scope of our work. It will certainly be interesting to explore this point in the future.
- It would be good to also measure the dry-weight of the seedlings (coleoptiles and roots separately) to explore the effect on the dry weight.
R: we thank the reviewer for this observation. Indeed we measured the dry weight of the plant material. Usually this data are not very common for molecular biology and for gene expression studies. However, since it could be of help for the general discussion, we re-calculated ROS and HPLC data based on the dry weight.
- With regard to the further experiments on the seedling materials it is clear how much plant materials is used (in grams), but not from how many seedlings this was taken as well as from how many treatments and if there was selection of seedlings on size for the samples taken?
R: we thank the reviewer for this remark. In the revised text we added all this information.
- With regard to the issue of possible delayed growth a lot could be learned from taking seedlings of similar size from both conditions (control and NNMF) to do the follow up tests (most importantly the H+-ATPase activity and expression level test). E.g. do NNMF seedlings with similar size as control (earlier harvested) show the same or different H+ATPAS activity?
R: this is an interesting remarks, but at the moment it is out of the scope of the work. Definitely we will keep this advice for further investigations.
- I expect to see also a MF dose-response curve for seedling growth.
R: the dose-response of MF was not the scope of this work. We already demonstrated that different MF intensities prompt a differential response in some genes (see for instance https://doi.org/10.1038/s41598-021-88695-6). The aim of this work was to show that the PM H+-ATPase and the 14-3-3 proteins are affected by the NNMF conditions, which are typical for example of the deep space. Further studies will investigate dose-responses and many other aspects.
- Why refer the activity and expression measurements to the fresh weight? Is reference to the dry weight not more logical?
R: see the answer to point 6
- It is confusing to show effects smaller than 1 (or -1) as 1 or -1
R: we are sorry for this misunderstanding. We decided to change the format of data representation from figures to Tables, by keeping the original values.
- The discussion is a bit too long. Could be reduced by 30%.
R: thanks for this remark. The discussion has been reduced accordingly.
Minor issues:
Lines 132-133: could be result of both (and/or instead of or).
R: done
Figures 4 A and B: show on the y-axis what is shown (e.g. GF14-6 etc).
R: done
Line 154: “As a result ….” This is not per se a result
R: the sentence has been rephrased
Line 154: “of 14-3-3 in “ should be “of 14-3-3 proteins in “
R: done
Line 221: “gene expression of ROS”. What is that??
R: the sentence has been rephrased
Line 347: “if both ….” Should be “of both …”
R: done
Reviewer 3 Report
This paper explores an interesting facet of plant biology, particularly the role of 14-3-3 proteins and plasma membrane H+-ATPase in the response of maize seedlings to changes in the geomagnetic field (GMF). It is clear that a significant amount of work and detailed experimentation has gone into this study. However, there are several areas of concern that need to be addressed before this paper can be considered for publication. Consequently, I suggest a major revision.
Major Comments:
-In the abstract, it is mentioned that transcriptomics were assessed, but no transcriptomic analysis (e.g., RNAseq) appears to be presented in the manuscript. This discrepancy needs to be resolved.
-The introduction is comprehensive but could benefit from being more concise to improve readability and focus on this specific study's aims.
-Some areas of the results section seem repetitive, especially in the explanations of the figures, which could be streamlined for brevity and clarity.
-Several conclusions drawn from the data appear overstated and could benefit from more cautious language. For example, the claim that "14-3-3 proteins and H+-ATPase require the GMF" should be tempered to "our results suggest..." or "our results indicate...".
-The presentation of qPCR results could be improved. The current linear scale makes interpretation difficult, and a logarithmic scale could be more appropriate. Also, the numerous qPCR figures could be combined into a heatmap for more efficient visualization.
-The discussion section could use careful editing to remove or soften language that overstates the conclusions.
-Speculative connections, such as linking observed changes in hormone levels to alteration of sucrose synthase gene expression, need to be toned down or supported by further data.
-The ROS-related data lacks clarity on whether all the measures (H2O2, gene expression changes) were performed in the same samples or independently. This needs to be clarified.
Author Response
We thank the reviewer for carefully reading our manuscript and for providing very useful comments that helped us to improve the revised text. Below we listed point-to-point answers to the reviewer's comments:
-In the abstract, it is mentioned that transcriptomics were assessed, but no transcriptomic analysis (e.g., RNAseq) appears to be presented in the manuscript. This discrepancy needs to be resolved.
R: we thank the reviewer for this remark. Gene expression has been used instead of transcriptomics
-The introduction is comprehensive but could benefit from being more concise to improve readability and focus on this specific study's aims.
R: introduction has been revised accordingly
-Some areas of the results section seem repetitive, especially in the explanations of the figures, which could be streamlined for brevity and clarity.
R: repetitions have been removed and the figure legends have been simplified.
-Several conclusions drawn from the data appear overstated and could benefit from more cautious language. For example, the claim that "14-3-3 proteins and H+-ATPase require the GMF" should be tempered to "our results suggest..." or "our results indicate...".
R: straight sentences have been tempered and rephrased as suggested
-The presentation of qPCR results could be improved. The current linear scale makes interpretation difficult, and a logarithmic scale could be more appropriate. Also, the numerous qPCR figures could be combined intoR: a heatmap for more efficient visualization.
R: By considering the comments of this reviewer and reviewer 2, we decided to represent data in tables and a general heatmap clustering for the visualization of gene expression results has been added.
-The discussion section could use careful editing to remove or soften language that overstates the conclusions.
R: straight sentences have been tempered and rephrased as suggested.
-Speculative connections, such as linking observed changes in hormone levels to alteration of sucrose synthase gene expression, need to be toned down or supported by further data.
R: the sentence has been revised and rephrased.
-The ROS-related data lacks clarity on whether all the measures (H2O2, gene expression changes) were performed in the same samples or independently. This needs to be clarified.
R: H2O2 has been measured on the same material used for gene expression. A new sentence clarifies this important issue in the materials and methods.
Round 2
Reviewer 2 Report
The authors significantly improved the manuscript and addressed adequately the majority of the issues raised in the first reviewed.
I still have a request with regard to my comment number 8 (and also 5). I fully understand that including data related to seedlings of the same size from control and NNMF conditions would require new experiments and analysis, which is a lot of work. I agree that such studies should be part of a follow up project. Nevertheless, it is important to realise that the effects found on H+-ATPases, 14-3-3 proteins, gene expression etc still may be due to differences in developmental stage and/or growth speed and only secondary to another MF sensitive target that is the primary target and causes a delay in development and/or speed of growth.
I still think that the authors should include such possibilities in the discussion (i.e. discuss the causal relationship between MF (or NNMF) and the effects) and the experiment(s) that could/should be done to proof the causal relationships.
Author Response
We thank the reviewer for suggesting this important issue.
As suggested, a new sentence has been added in the conclusion.
Reviewer 3 Report
Dear Authors,
I have reviewed the revised version of your manuscript and am satisfied with the changes made in response to my original comments. I believe the manuscript has improved considerably in clarity, conciseness, and accuracy.
I appreciate you thoughtfully addressing all the points I raised. The revisions to the abstract, introduction, results, and discussion have strengthened the work overall. The improved presentation of the qPCR data also facilitates interpretation. I am glad to see you toned down the speculative conclusions and clarified methods per my recommendations.
Overall, I am impressed with the authors' willingness to revise the manuscript according to my suggestions. I believe the current version is solid, well-written, and ready for publication. I have no further changes or revisions to recommend.
Congratulations on doing a good job revising this manuscript. I look forward to seeing this valuable work published soon.
Author Response
We thank very much the reviewer for the useful advice during the review process and for helping us to improve our manuscript.